# Relationship between Emotional Intelligence, Generativity and Self-Efficacy in Secondary School Teachers

**David Aparisi [1],\*, Lucía Granados [2], Ricardo Sanmartín [1]**,
**María Carmen Martínez-Monteagudo [1] and José Manuel García-Fernández [1]**

[1] Department of Developmental Psychology and Teaching, University of Alicante, 03080 Alicante, Spain; ricardo.sanmartin@ua.es (R.S.); maricarmen.martinez@ua.es (M.C.M.-M.); josemagf@ua.es (J.M.G.-F.)

[2] Department of Education, International University of Valencia, 46002 Valencia, Spain; lucia.granados@campusviu.es

\* Correspondence: david.aparisi@ua.es; Tel.: +34-9659-034-00

**Abstract:** The aim of this study was to examine the relationship between emotional intelligence (EI), generativity and self-efficacy, identifying different profiles of emotional intelligence. 834 secondary school teachers participated in the study by completing the Trait Meta-Mood Scale–24 (TMMS–24), the Loyola Generativity Scale and the General Self-Efficacy Scale. Cluster analysis identified four EI profiles: the first with high scores in attention and low scores in repair, the second with high scores in all dimensions of EI, the third with low scores in all EI dimensions and a fourth profile with low scores in attention and high scores in repair. Results showed significant statistical differences between the EI profiles found and the different dimensions of generativity and self-efficacy. Logistic regression analysis showed that EI was a statistically significant predictor of generativity, since teachers with high EI scores were more likely to present high scores in positive generativity and self-efficacy and lower probability of presenting high scores in generative doubts.

**Keywords:** cluster analysis; emotional intelligence; generativity; logistic regression; self-efficacy

## 1. Introduction

Students spend most of the time in the classroom during their school years, and it is in this period of time that their emotional development is produced. Therefore, the school environment is a fundamental space of emotional socialisation, and teachers become references for attitudes, values, behaviours, feelings and emotions [1]. There is no doubt about the role that teachers as educational agents play in supporting the cognitive and emotional needs of children in their learning process [2].

In addition to this emotional aspect, teachers have the duty of guiding and taking care of new generations, and this worry is the central component of generativity. In this sense, the basic assumption of generative ethics consists in performing actions that will promote both other people's and one's own development [3]. There are personal traits of teachers, such as trust, support and emotional intelligence, that are key to facilitating the generativity of knowledge for students [4].

Related to the previous idea, another variable appears which is especially important and decisively influences the teachers' performance: self-efficacy. This variable is understood as the assessment that each individual makes of their own capacities and used as basis for organising and carrying out actions in order to achieve an adequate performance [5]. Consequently, it can be said that an adequate teacher performance depends on these abilities, on the one hand, and on the belief in their own capacity, on the other.

The main aim of this study is to examine the relation between emotional intelligence (EI), generativity and self-efficacy in a sample of secondary education teachers.

## 2. Theoretical Background

### 2.1. Emotional Intelligence in Teachers

Mayer and Salovey [6] define EI as "the ability to perceive, assess and express emotions with accuracy, to access and/or generating feelings that facilitate thinking; to understand emotions and emotional knowledge and regulating emotions through an intellectual and emotional growth". These authors formulate a model with four branches, in which EI has several dimensions or abilities [7]: (a) *Attention*: it is defined as the ability to identify and recognise both one's own and other people's feelings, (b) *Clarity*: it makes reference to the ability to label emotions and recognise in which categories feelings are grouped in, and (c) *Repair*: it is the ability to regulate one's own emotions and the emotions of others, moderating the negative emotions and intensifying the positive ones.

It is important to mention as well that there are other models that conceptualise EI as a stable personality trait [8]. This study is based on the idea that EI is an ability which can be learnt and trained.

Several studies have related EI to the teaching profession, because teachers have to make use of their EI during their activity to successfully guide their own emotions and the students' emotions [9].

A recent research with university students had the objective of examining the relations between perceived social support, EI and academic performance. Results showed higher associations between teacher support and academic performance, family support and emotional repair, and emotional repair and academic performance. Similarly, the explanatory capacity of teacher support was favoured over academic performance, family support over emotional clarity and emotional repair, and emotional repair over academic performance [10].

Another piece of research aimed to analyse the relation between EI, academic commitment and academic performance in a sample of 3512 secondary school students. Results showed a positive relation between EI, academic commitment and academic performance [11].

### 2.2. Impact of Generativity on School Environment

Certain professions, such as teaching, are irretrievably generative. In fact, one of the aims of the education system is the education of the following generations [12]. Generativity is closely related to intrinsic motivation [13], because the more motivated a person is to engage in close experiences and exchanges with other people, and to influence other people's nature and emotions, the higher the probability that he or she will generate cross-generational objectives in the future [14].

Teachers are a collective group that presents a specific culture, which is transmitted to the students through education. In the words of López and Pérez [15], teaching is the set of beliefs, values, habits and prevailing rules that determine the aspects considered as valuable for the professional context, as well as the politically correct ways of thinking, felling, acting and interacting with each other. A positive assessment and a certain ethical obligation of cross-generational transition take part in these beliefs.

As a result, it can be said that generativity is placed in the centre of teaching culture, because when educational practise is satisfactory, it benefits both parties involved. On the one hand, students are benefited by a perceived support, and on the other hand, teachers are benefited by guiding the next generation and feeling needed [16].

In this way, generativity could have great importance as a predictor of personal well-being based on teaching [17]. Thus, the generative process acts as a preventive factor of burnout syndrome because it helps teachers to interpret their motivations and goals in a wider context, which is line with the teaching role [18]. However, when expectations of interest, commitment and generative action are very high in teaching professionals, burnout syndrome or teachers' malaise may appear [19]. In this sense, the lack of generativity could be considered as an evolutionary predecessor of burnout [18].

In the case of teachers, a study performed with a sample of primary and secondary education teachers showed that secondary education teachers obtained significantly lower scores in generativity, and it was not possible to explain whether the discouragement was a consequence of a generative worry due to lack of opportunities or simply a lack of initial motivation. Conversely, generative professors showed higher scores in personal realisation in comparison with other stalled teachers [20].

In the case of students, a piece of research done with university students showed that students considered to be tutors, who actively collaborate with the next generation, demonstrated a significantly higher generativity than students without the role of tutor [21].

In fact, various authors [22] argue for a (re)new(ed) emphasis on community-based teacher preparation grounded in critical reflection and generativity, which facilitates and promotes transformative teacher education that prepares teachers to teach diverse student populations.

### 2.3. Self-Efficacy and Its Relation with the Teaching Activity

Self-efficacy is defined by Schwarzer and Schmitz [23] as the confidence in one's coping skill, which is manifested in a wide range of defiant situations, and it has a stable and comprehensive character.

In the work environment, self-efficacy regulates the relation between stressors such as working hours, overload and the consequences of these facts for the individual, for example the satisfaction or dissatisfaction at work, physical symptoms, the intention of abandoning the workstation and the commitment to the organisation [24]. In this sense, self-efficacy is also related to burnout syndrome because the experience of stress is the result of low efficacy when controlling a stressful situation [25,26].

Several studies have related self-efficacy with the teaching activity. A study with 467 secondary education teachers from China analysed the relation between teachers' EI and self-efficacy, and it also assessed whether this relation was mediated by teaching performance. The results showed that high levels of EI positively correlated with high levels of self-efficacy. This relation was partially mediated by teaching performance [27].

Another study recently carried out with 350 secondary education professors analysed the self-efficacy related with information and communication technologies (ICT), innovation support and emotions between teachers. The results showed that self-efficacy and support predicted positive emotions towards ICT and company satisfaction. Besides, these variables positively predicted self-motivation and working commitment [28].

### 2.4. The Current Study

Although previous empirical research has tested the relation between EI and diverse educational variables, such as the academic [29], personal [30] and social performance [31], there is a lack of studies examining the concrete relation between EI dimensions, generativity and the self-efficacy of secondary education teachers. As a result, the aim of this study was threefold: (1) to identify different profiles of EI considering its dimensions (attention, clarity and emotional repair), (2) to test the existence of statistically significant differences in the scores of generativity and self-efficacy according to the EI profiles identified, and (3) to analyse the predictive capacity of EI over the dimensions of generativity and self-efficacy in a sample of secondary education teachers.

From the revision of previous studies, we expected to find differences between the EI profiles in the generativity and self-efficacy scores. Concretely, the following hypotheses were raised: (1) it is expected that teachers with profiles based on low general EI and high scores in attention and low in clarity and repair will obtain higher scores in generative doubts and lower scores in positive generative; (2) it is expected that teachers with these profiles (low general EI and high scores in attention and low in clarity and repair) will present significantly lower scores in self-efficacy; (3) it is expected that EI will be a statistically significant predictor variable of generativity; and at the same time, (4) it is expected that EI will be a statistically significant predictor variable of self-efficacy in the current sample.

## 3. Materials and Methods

### 3.1. Participants

A random sampling by clusters was carried out (geographic zones: Murcia region, Aragón and Alicante province—centre, north, south, east and west). With the aim of representing all the geographic zones, between 1 and 6 centres per zone were randomly chosen according to the population, obtaining a total of 30 centres from rural and urban areas.

Once the centres were selected, school leadership teams were contacted by phone or email. The aims of the study were explained, and it was requested that the information be provided to the teachers, in order for them to answer the online questionnaires and participate in the study.

The total sample was formed by 834 teachers (476 men and 358 women), from a total of 1080, because 246 teachers (22.8%) did not participate in the research due to mistakes or omissions in their answers. The mean age of the participants was between 29 and 65 years ($M = 45.81$; $SD = 13.35$), and the ethnic composition was the following: 94.9% Spanish and 5.1% from the rest of Europe.

Using the Chi-square Test of Homogeneity of frequencies distribution, it was proved that there were no statistically significant differences among the eight groups of Sex x Course ($\chi^2 = 3.15$; $p = 0.37$).

### 3.2. Instruments

#### 3.2.1. Trait Meta-Mood Scale (TMMS–24)

This instrument [7] is a Spanish adaptation of the TMMS–48 elaborated by Salovey, Mayer, Goldman, Turvey and Palfai [32]. This adaptation is formed by 24 items of a 5-point Likert scale (1 = *completely disagree*; 5 = *totally agree*). The items are distributed in three scales: (1) Emotional attention: referred to the ability to identify one's own emotions and other people's emotions, through attention and decoding of verbal and non-verbal correlations (e.g., "I always can say how I feel"), (2) Emotional clarity: understood as the ability to reflect about emotional information, understanding the relations between emotions, their simultaneity and their blending, as well as their progressions and transitions over time (e.g., "I can come to understand my feelings") and (3) Emotional repair: referred to the capacity of being open to emotional states, both positive and negative, reflecting about their utility and informative value, as well as controlling personal and alien emotions, moderating the negative emotions and keeping the positive, without suppressing or exaggerating the information that they provide (e.g., "When I am angry, I try to change my mood").

The reliability ($\alpha$) of this instrument in its Spanish version [7] (Fernández-Berrocal et al. 2004) was 0.90, 0.90 and 0.96 for the scales of Attention, Clarity and Repair, respectively. The internal consistency for the current study was 0.86 for the Attention scale, 0.83 for the Clarity scale and 0.83 for the Emotional Repair scale.

#### 3.2.2. Loyola Generativity Scale (LGS)

This is a self-report questionnaire [33] composed of 20 items (e.g., "I try to transmit to others the knowledge that I have learnt through my own experiences") which measures the generativity understood as the concern with guiding the new generations. The original version of this scale shows a high internal consistency (Cronbach's Alpha 0.82 and 0.83) and adequate test-retest reliability (0.73) at three weeks. In the original version of LGS, scores higher than 45 points express generative concern and responsibility consciousness with regard to young members of society, whereas scores lower than 10 points denote a self-image with low capacity to influence other people, low interest to share experiences and knowledges with other people and lack of necessity of guiding the next generation. The LGS appears to be a unidimensional scale; an exploratory factor analysis found that the LGS loaded on two factors, but these were distinguished only by question wording, as one factor included positively worded items and another included negative, reverse-coded items [34].

### 3.2.3. General Self-Efficacy Scale (GSE)

It is an instrument [35] directed to assess the stable feeling of personal competence to effectively manage a variety of stressful situations. It is formed by 10 items (e.g., "Thanks to my qualities I can overcome unforeseen situations") in a 4-points Likert scale, with a direct score ranging between 1 and 40 points (the high scores are related to high self-efficacy perception). The scale shows a consistency value of 0.93 and reliability values between 0.92 and 0.93. This questionnaire was developed by Jerusalem and Schwarzer [36], and it was adapted to several languages (German, French, Spanish and Portuguese), evidencing good psychometric properties.

### 3.3. Procedure

The study design corresponds to the quantitative type with an experimental research design [37]. Firstly, once the centres were selected, a meeting with the head teachers was held with the aim of presenting the objectives of the current study and the assessment measures used, requesting their consent and promoting their collaboration in order to facilitate the information to the teachers that had to fill in the questionnaires.

Subsequently, an information letter for all the teachers of the centre was elaborated, in which the aim of the study was explained and their collaboration with the investigation was requested. Once all the aspects were defined, the questionnaires were administered. The measures were administered online, and their fulfilment was voluntary, individual and anonymous. The total fulfilment of the instruments was emphasised and the average time of fulfilment was 15 min. Additionally, the answers were treated according to the ethical principles of anonymity and confidentiality prescribed in the Declaration of Helsinki.

### 3.4. Statistical Analyses

To identify the EI profiles, the cluster analysis called quick cluster analysis was used. This cluster analysis is not only the most appropriate procedure to establish profiles in a broad sample of subjects [38] but also one of the most recommended solutions to identify multiple goals [39].

The profiles have been defined according to the combination of the three factors of EI which are assessed by the TMMS–24: emotional attention, emotional clarity and emotional repair. The criterion followed for choosing the number of clusters was the maximisation of the inter-cluster differences, with the aim of achieving the largest number of possible groups with different combinations of the EI dimensions. Moreover, the theoretic viability and psychological significance of each of the groups that represented the different EI profiles were also added to the criteria. After establishing the different groups through cluster analysis, analyses of variance (ANOVA) were performed in order to analyse the statistical significance of the existent differences between the groups according to the generativity and self-efficacy dimensions. To know the magnitude or effect size of the mentioned differences, the eta-squared index was obtained. Afterwards, when statistically significant differences were found, post hoc tests were performed to identify the groups that caused the differences. The Scheffé test was used because it does not need equal sample sizes. Besides, the $d$ effect size [40] was calculated to obtain the magnitude of the fond differences. Its interpretation is the following: a small effect size is located between $0.20 \leq d \leq 0.49$, moderate between $0.50 \leq d \leq 0.79$ and large $d \geq 0.80$. Finally, the analyses were carried out with the statistical package SPSS version 20.

The establishment of predictive equations for EI was performed with the statistical technique called logistic regression, following the stepwise regression procedure based on the Wald test. During the logistic regression analysis, the coefficients of each variable were presented in the regression equation and the achieved statistics in the models where individuals were classified according to the group to which they belong. The logistic modelling allows researchers to estimate the probability of an event, incident or result occurring (e.g., high EI), versus not occurring, in presence of one or more predictors (e.g., high scores in self-efficacy). This probability is estimated with the Odd ratio

(OR) estimator, whose interpretation is the following: if OR is higher than one, for instance 3, for each time the event occurs in presence of the independent variable, it will occur three times if the variable is present. On the contrary, if OR is lower than one, for instance 0.5, the probability of an event happening in the absence of the independent variable will be higher than in its presence. The quality of the proposed models and the adjustment was assessed with Nagelkerke's $R^2$, which indicates the percentage of variance explained by the model [41] and the percentage of cases correctly classified by the model or their predicted efficiency.

## 4. Results

### 4.1. Emotional Intelligence Profiles

The cluster analysis allowed us to identify four EI profiles according to the combination of its three dimensions (Attention, Clarity and Emotional Repair). As Figure 1 shows, the first profile was formed by 281 teachers (33.7%) who presented high scores in attention and low scores in repair (HALR). The second profile was formed by 206 teachers (24.7%) who presented high scores in all the EI dimensions. As a result, they showed high general EI (HGEI). The third profile was formed by 122 teachers (14.6%) who presented low scores in all the EI dimensions. As a result, they presented low general EI (LGEI). A fourth profile was formed by 225 teachers (27%) who presented low scores in attention and high scores in repair (LAHR).

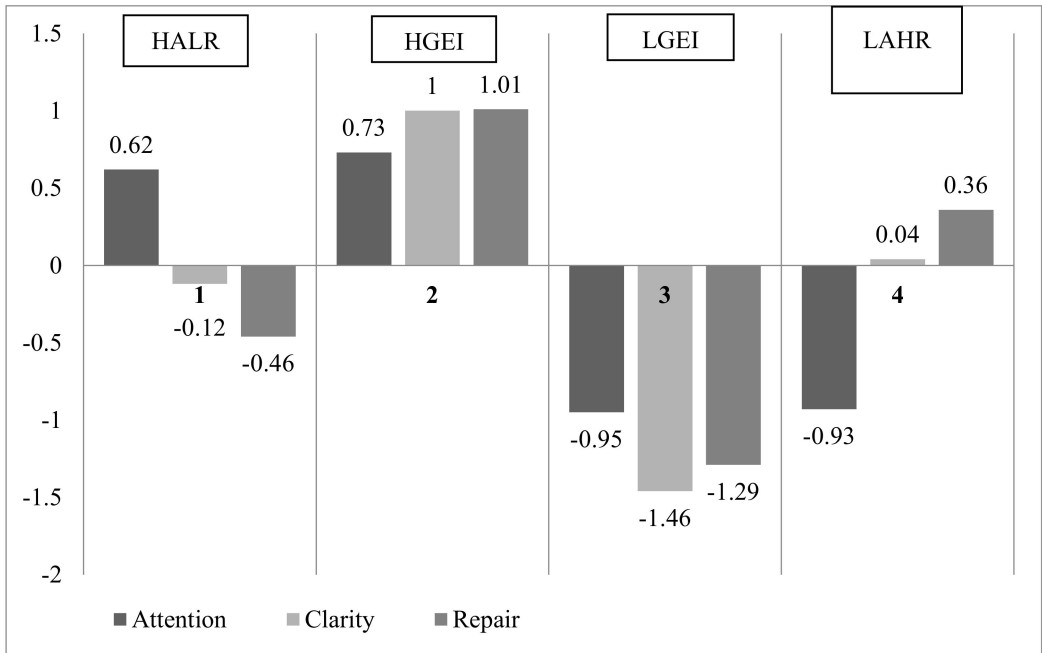

**Figure 1.** Emotional intelligence profiles through cluster analysis. Note: HALR (High Attention and Low Repair), HGEI (High General Emotional Intelligence), LGEI (Low General Emotional Intelligence) and LAHR (Low Attention and High Repair).

### 4.2. Inter-Group Differences in the Generativity and Self-Efficacy Dimensions

The ANOVA results (see Table 1) showed the existence of statistically significant differences in the dimensions of positive generativity ($p < 0.001$), generative doubts ($p < 0.001$) and self-efficacy ($p < 0.001$).

**Table 1.** Means and standard deviations obtained in the four profiles for each dimension of generativity and self-efficacy.

| | Group 1 HALR | | Group 2 HGEI | | Group 3 LGEI | | Group 4 LAHR | | Statistical Signification | | |
|---|---|---|---|---|---|---|---|---|---|---|---|
| **Dimensions** | *M* | *SD* | *M* | *SD* | *M* | *SD* | *M* | *SD* | $F_{(3830)}$ | $\eta^2$ | *p* |
| Positive Generativity | 14.49 | 3.12 | 16.62 | 4.03 | 12.17 | 3.75 | 14.65 | 3.63 | 40.10 | 0.13 | <0.001 |
| Generative Doubts | 5.89 | 1.69 | 6.31 | 1.87 | 7.51 | 1.84 | 6.22 | 1.75 | 23.98 | 0.08 | <0.001 |
| Self-Efficacy | 73.66 | 8.32 | 82.91 | 11.04 | 70.66 | 11.22 | 79.09 | 9.79 | 55.29 | 0.18 | <0.001 |

Note: HALR (High Attention and Low Repair), HGEI (High General Emotional Intelligence), LGEI (Low General Emotional Intelligence) and LAHR (Low Attention and High Repair).

With regard to the positive generativity dimension, the post hoc contrasts indicated that teachers with high scores in attention and low scores in repair (HALR) and teachers with low general EI (LGEI) obtained significantly lower scores than teachers with high general EI (HGEI), with moderate effect sizes in both cases (*d* = 0.60 and *d* = 0.70, respectively). On the other hand, teachers with high general EI (HGEI) obtained significantly higher scores than teachers with low general EI (LGEI) and teachers with a profile characterised by low scores in attention and high in repair (LAHR), with large (*d* = 1.13) and moderate (*d* = 0.51 and *d* = 0.68) effect sizes, respectively. Finally, teachers with a profile of low general EI (LGEI) obtained significantly lower scores than teachers with low scores in attention and high scores in repair (LAHR), with a moderate effect size (*d* = 0.68).

With reference to the dimension of generative doubts, teachers with low general EI (LGEI) obtained significantly higher scores than the group of teachers with high scores in attention and low in repair (HALR), the group of teachers with high general EI (HGEI) and the group of teachers with low scores in attention and high scores in repair (LAHR), with very large (*d* = 0.93) and moderate (*d* = 0.65, *d* = 0.72) effect sizes, respectively.

Concerning the self-efficacy dimension, teachers with high general EI (HGEI) obtained significantly higher scores than the group of teachers with high scores in attention and low in repair (HALR), teachers with low general EI (LGEI) and the group of teachers with low scores in attention and high in repair (LAHR), with very large (*d* = 0.97, *d* = 1.10) and low (*d* = 0.37) effect sizes, respectively. On the other hand, teachers with low scores in attention and high in repair (LAHR) obtained significantly higher scores in comparison with teachers with high scores in attention and low in repair (HALR), with a moderate effect size (*d* = 0.60). Finally, teachers with low general EI (LGEI) obtained significantly lower scores than teachers with low scores in attention and high in repair (LAHR), with a large effect size (*d* = 0.82).

### 4.3. Predictive Capacity of Emotional Intelligence over Generativity and Self-Efficacy

The binary logistic regression analyses showed that EI was a statistically significant variable to predict both the generativity and self-efficacy dimensions.

From the analysed sample it was possible to create three predictive models: one for the prediction of positive generativity, one for the prediction of generative doubts and another for the prediction of self-efficacy. The EI dimensions (attention, clarity and repair) were included as predictor variables in all the logistic models created.

The model for positive generativity correctly classified 69.8% of the cases ($\chi^2$ = 96.87, *p* < 0.001) for the different EI dimensions (Nagelkerke's $R^2$ = 0.21).

The OR of the logistic models for the prediction of positive generativity showed that (see Table 2): (1) teachers with high emotional attention were 5% more likely to present high positive generativity, (2) teachers with high emotional clarity were 8% more likely to present high positive generativity, and (3) teachers with high emotional repair were 9% more likely to present high positive generativity.

**Table 2.** Binary logistic regression for the probability of presenting high scores in positive generativity and in generative doubts according to emotional intelligence.

| DV | IV | B | S.E. | Wald | *p* | OR | C.I. 95% |
|---|---|---|---|---|---|---|---|
| Positive Generativity | *Attention* | 0.05 | 0.02 | 10.12 | 0.001 | 1.05 | 1.02–1.09 |
| | *Clarity* | 0.07 | 0.02 | 13.79 | 0.001 | 1.08 | 1.03–1.12 |
| | *Repair* | 0.08 | 0.02 | 17.61 | 0.001 | 1.09 | 1.04–1.13 |
| | Constant | −5.78 | 0.73 | 63.34 | 0.001 | 0.003 | |
| Generative Doubts | *Attention* | −0.061 | 0.021 | 8.03 | 0.005 | 0.94 | 0.90–0.98 |
| | *Clarity* | −0.055 | 0.021 | 7.11 | 0.008 | 0.95 | 0.91 0.99 |
| | Constant | 3.79 | 0.720 | 27.85 | 0.001 | 44.58 | |

Note: A = Attention, C = Clarity, R = Repair; B = Regression coefficient; S.T. = Standard Error; Wald = Wald's test; *p* = Probability; *OR* = Odd ratio; C.I. = Confidence interval at 95%.

Moreover, the model of generative doubts correctly classified 57.9% of the cases ($\chi^2$ = 25.68, $p < 0.001$) for the EI dimensions (Nagelkerke's $R^2$ = 0.10).

The OR of the logistic models for the prediction of generative doubts showed that (see Table 2): (1) teachers with high emotional attention were 6% less likely to present high scores of generative doubts, and (2) teachers with high emotional clarity were 5% less likely to present high scores in generative doubts.

Finally, the model of self-efficacy correctly classified 75.7% of the cases ($\chi^2$ = 168.58, $p < 0.001$) for the EI dimensions (Nagelkerke's $R^2$ = 0.44).

The OR of the logistic models for the prediction of self-efficacy showed that (see Table 3): (1) teachers with high emotional attention were 12% less likely to present high scores of self-efficacy, (2) teachers with high emotional clarity were 15% more likely to present high scores of self-efficacy, and (3) teachers with high emotional repair were 21% more likely to present high self-efficacy scores.

**Table 3.** Binary logistic regression for the probability of presenting high scores in self-efficacy according to emotional intelligence.

| DV | IV | B | S.E. | Wald | *p* | OR | C.I. 95% |
|---|---|---|---|---|---|---|---|
| Self-Efficacy | *Attention* | −0.13 | 0.03 | 26.97 | 0.001 | 0.88 | 0.84–0.92 |
| | *Clarity* | 0.14 | 0.03 | 24.80 | 0.001 | 1.15 | 1.09–1.21 |
| | *Repair* | 0.19 | 0.03 | 47.62 | 0.001 | 1.21 | 1.15–1.28 |
| | Constant | −6.59 | 0.91 | 51.98 | 0.001 | 0.01 | |

Note: A = Attention, C = Clarity, R = Repair; B = Regression coefficient; S.T. = Standard Error; Wald = Wald's test; *p* = Probability; *OR* = Odd ratio; C.I. = Confidence interval at 95%.

## 5. Discussion

The general aim of this study was to analyse the relation between EI, generativity and self-efficacy in a sample of secondary education teachers.

Unlike the previous research, this study analyses the importance of generativity and self-efficacy in the school environment from the analysis of different profiles of teachers considering EI. Moreover, this piece of research has considered the mentioned relation taking into account the effect sizes, which are recommended by several authors to determine the magnitude of the differences found (e.g., [39,42]).

The first specific objective of this study was to identify different EI profiles according to their dimensions (attention, clarity and emotional repair). The cluster analysis allowed us to identify four EI profiles: a first profile with high scores in attention and low scores in repair (HALR), a second profile with high scores in all the EI dimensions (and, as a result, with the individuals presenting a high level of general EI (HGEI)), a third profile with low scores in all the EI dimensions (and, consequently, with the individuals presenting low levels of general EI (LGEI)) and a fourth profile with low scores in attention and high scores in emotional repair (LAHR). These profiles are in line with the results obtained by previous investigations [43–45].

With regard to the second specific objective, results show statistically significant differences between the EI profiles in the different dimensions of generativity and self-efficacy. Concretely, teachers with profiles of HALR and LGEI presented significantly lower scores in positive generativity, and only the profile of teachers with LGEI presented significantly higher scores in generative doubts, partially confirming the first hypothesis. On the other hand, the same profiles of teachers with HALR and LGEI presented significantly lower scores in self-efficacy, confirming the second hypothesis of the study. These results support the idea of the existence of a relation between EI and generativity and its impact on life satisfaction [46], as well as between EI and self-efficacy, both in teachers [47] and students [48].

With regard to the third objective, it was proved that EI was a statistically significant predictor variable for generativity, because teachers with high scores in the different dimensions (attention, clarity and repair) presented a higher probability of scoring high in positive generativity and a lower probability of scoring high in generative doubts, confirming the third hypothesis. Besides, it was also shown that EI was a statistically significant predictor variable of self-efficacy, because teachers with high scores in the emotional clarity and repair dimensions of EI presented a higher probability of obtaining high scores in self-efficacy, confirming the fourth hypothesis. These findings reveal the synergy between EI, which is very necessary in the school environment, and generativity, which focuses on contributing to others' well-being, in this case students [49].

## 6. Conclusions

In conclusion, this piece of research confirms the existence of different EI profiles according to the different dimensions of the construct (attention, clarity and repair). Concretely, four EI profiles were found in the analysed teachers sample: a first profile with high scores in attention and low scores in repair (HALR), a second profile with high scores in all the EI dimensions (and, as a result, with the individuals presenting a high level of general EI (HGEI)), a third profile with low scores in all the EI dimensions (and, consequently, with the individuals presenting low levels of general EI (LGEI)) and a fourth profile with low scores in attention and high scores in repair (LAHR). On the other hand, statistically significant differences were found between the different EI profiles and the dimensions of generativity and self-efficacy. Concretely, teachers with profiles of HALR and LGEI presented significantly lower scores in positive generativity, the profile of teachers with LGEI presented significantly higher scores in generative doubts, and the same two profiles of teachers with HALR and LGEI presented statistically significant lower scores in self-efficacy. Finally, it was observed that EI acted as a statistically significant predictor variable of the different dimensions of generativity and self-efficacy, because teachers with high scores in all the different dimensions of EI (attention, clarity and repair) presented more probability of scoring high in positive generativity and self-efficacy and a lower probability of scoring high in generative doubts

## 7. Limitations and Future Research Lines

This study presents some limitations. Firstly, although the sampling method used guarantees the representativeness of the sample, the results found in this study cannot be generalised to teachers of other educational levels. Future investigations should confirm whether these results on secondary education can be replicated in other educational levels. Additionally, it would be advisable for future works to use longitudinal designs to provide more conclusive data with regard to the influence relations between these variables. Finally, the current study aims to determine the predictive capacity of EI over generativity and self-efficacy, and not the other way round (the predictive capacity of generativity and self-efficacy over EI). Although the logical would be to think about a reciprocal effect, future investigations could analyse this fact performing two models of structural equations to test which hypothesis is more robust or which association force appears between both models.

On a practical level, the results of the current study point, in the first place, to the support of the effectiveness of programmes aimed to enhance EI levels [50]. Moreover, an increase of EI improves, to a large extent, the teachers' self-efficacy, because when emotional abilities are used with success,

this fact provokes a better teacher performance [27], and teachers become a solid social support for students [51].

In the second place, generativity can act as a protecting factor for teachers, because the more generative teachers present high levels of motivation and personal fulfilment and show a higher grade of personal well-being, especially middle-aged teachers [52]. It can be expected that more generative teachers are better protected against burnout, because their concern for the students' development and their commitment to guiding them, teaching them and imparting values, rules and adequate models to them, all of which are inherent to generativity, are intrinsic to the teachers' motivations [53]. The study of generativity in the school environment offers important contributions to leadership and social responsibility [21].

Finally, self-efficacy beliefs can cushion the impact of stressors on people through the implementation of adequate coping measures [54], and they can be used as a mediator between EI, teaching abilities and academic performance [55].

Therefore, this study has important repercussions for educational practice, since it shows the need to promote interventions aimed at developing the emotional skills of teachers and fostering their sense of self-efficacy, which directly affects their well-being, and the ability to positively influence the next generations, on the one hand, and to create a better climate for the students, on the other.

**Author Contributions:** All authors have read and agree to the published version of the manuscript. D.A. and J.M.G.-F. conceived and designed the methodology; J.M.G.-F. and L.G. were in charge of the resources, calculation and investigation of the data; D.A. and R.S. were responsible for writing—review and editing, and M.C.M.-M. for the supervision.

**Funding:** No grant was obtained for this study.

**Acknowledgments:** The authors thank the educational institutions and the teachers for their voluntary participation in this study.

**Conflicts of Interest:** The authors declare that they have no conflicts of interest.

**Research Involving Human Participants:** The present study was a part of the doctoral research of the second author.

**Informed Consent:** Informed consent was obtained from all individual participants included in the study.

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
