# Peer review of "Relationship between Emotional Intelligence, Generativity and Self-Efficacy in Secondary School Teachers"

_sustainability, doi:10.3390/su12103950_

Round 1

Reviewer 1 Report

Thank you for inviting me to review this manuscript, titled “Relationship between emotional intelligence, generativity and self-efficacy in secondary school teachers”. This study examines the relation between emotional intelligence, generativity and self-efficacy in a wide sample of secondary education teachers. The importance of generativity and self-efficacy is studied in the school environment from the analysis of different teacher profiles considering emotional intelligence.

The main question addressed by the research is relevant and interesting. The topic is original. The specific issue or problem is defined.

The title and abstract are structured according to the sections indicated in the journal, clearly specifying the objective, the method used, the results and conclusion.

The proposed objective is relevant, as well as the evidence provided and the accompanying discourse. It gathers coherent and argued information at the different key points of the process.

However, the manuscript would benefit from some small suggestions or changes. See specific suggestions below.

Sort the keywords in the abstract alphabetically.

Although the theoretical approach is correct, with an adequate and precise conceptualization of the construct that is being addressed; If possible, a more specific theoretical model could be added that integrates the study variables.

Check whether some statistics such as p in the in in participants on page 4 should be in italics.

It could indicate the type of design that has been used in the study.

On page 2, when this textual information must appear, in addition to the authors' last names and the year, the page number: “Mayer and Salovey (1997) define EI as “the ability to perceive, assess and express emotions with 43 accuracy, to access and/or generating feelings that facilitate thinking; to understand emotions and 44 emotional knowledge and regulating emotions through an intellectual and emotional growth”

In the legend of figure 1 it is not necessary to put “Graphic representation…” but it can be put directly “Figure 1. Emotional intelligence profiles through cluster analysis”

Check whether on page 9 there has been any type of error in the following references: “Authors, 2015; Authora, 2017)”

Some reference on the new page is not in the same font size: (van Laren, Mudaly, Pithouse-Morgan and Singh, 2013)

Please check if the reference is “…(Antonio-Aguirre, Rodríguez,Fernández and Revuelta, 2009)” from the year 2009 and it is necessary to add it in the final list of references or as it appears in the final list of references is from the year 2019 and it is necessary to change the year of said reference within the text in the page 10

In the final reference list there are some typographical errors: Kotsou, I., Mikolajczak, M., Heeren, A., Gregoire, J., & Leys, C. (2019). I

More suggestions or proposals for future research could also be presented after the literature consulted and especially a paragraph that indicates the importance and psychoeducational implications derived from the study.

Author Response

Reviewer 1

Firstly we would like to thank you and the Reviewer for the positive overall evaluation of our manuscript and for your constructive comments, which have allowed us to improve the quality of the work considerably.

Based on his comments, several changes have been made to the revised manuscript:

  1. Sort the keywords in the abstract alphabetically

Keywords have been ordered alphabetically.

  1. Although the theoretical approach is correct, with an adequate and precise conceptualization of the construct that is being addressed; If possible, a more specific theoretical model could be added that integrates the study variables.

Following the suggestion, the manuscript has been restructured.

  1. Check whether some statistics such as p in the in participants on page 4 should be in italics.

It has been modified in the text.

  1. It could indicate the type of design that has been used in the study.

The type of design has been indicated in the text.

  1. On page 2, when this textual information must appear, in addition to the authors' last names and the year, the page number: “Mayer and Salovey (1997) define EI as “the ability to perceive, assess and express emotions with 43 accuracy, to access and/or generating feelings that facilitate thinking; to understand emotions and 44 emotional knowledge and regulating emotions through an intellectual and emotional growth”

It has been modified in the reference.

  1. In the legend of figure 1 it is not necessary to put “Graphic representation…” but it can be put directly “Figure 1. Emotional intelligence profiles through cluster analysis”

It has been modified in the text.

  1. Check whether on page 9 there has been any type of error in the following references: “Authors, 2015; Authora, 2017)”

It has been modified in the references.

  1. Some reference on the new page is not in the same font size: (van Laren, Mudaly, Pithouse-Morgan and Singh, 2013)

It has been modified in the text.

  1. Please check if the reference is “…(Antonio-Aguirre, Rodríguez,Fernández and Revuelta, 2009)” from the year 2009 and it is necessary to add it in the final list of references or as it appears in the final list of references is from the year 2019 and it is necessary to change the year of said reference within the text in the page 10

It has been modified in the text.

  1. In the final reference list there are some typographical errors: Kotsou, I., Mikolajczak, M., Heeren, A., Gregoire, J., & Leys, C. (2019).

It has been modified in the references.

  1. More suggestions or proposals for future research could also be presented after the literature consulted and especially a paragraph that indicates the importance and psychoeducational implications derived from the study.

A paragraph has been added in the text describing some practical implications.

Reviewer 2 Report

The introduction section is too long. I think you have to resume the main concepts because there is too much content and it's not necessary.

Methods: you don't explain the study design, what it is very important. And it could be better to explain the ethics considerations of the study.

Good work

Author Response

Reviewer 2

Firstly we would like to thank you and the Reviewer for the positive overall evaluation of our manuscript and for your constructive comments, which have allowed us to improve the quality of the work considerably.

Based on his comments, several changes have been made to the revised manuscript:

  1. The introduction section is too long. I think you have to resume the main concepts because there is too much content and it's not necessary.

Thank you very much for your comment. Following the suggestion, the manuscript has been restructured so that the introduction is much shorter.

  1. Methods: you don't explain the study design, what it is very important. And it could be better to explain the ethics considerations of the study.

Design type and compliance with ethical standards have been added to the manuscript.
